# Inmates with Harmful Substance Use Increase Both Exercise and Nicotine Use Under Incarceration

**DOI:** 10.3390/ijerph15122663

**Published:** 2018-11-27

**Authors:** Ashley Elizabeth Muller, Ingrid Amalia Havnes, Eline Borger Rognli, Anne Bukten

**Affiliations:** 1Norwegian Centre for Addiction Research, Pb 1039 Blindern, 0135 Blindern, Norway; anne.bukten@medisin.uio.no; 2Institute of Clinical Medicine, University of Oslo, 0318 Oslo, Norway; 3National Advisory Unit on Substance Use Disorder Treatment, Oslo University Hospital, Pb 4959 Nydalen, 0424 Oslo, Norway; UXAMAV@ous-hf.no; 4Section for Clinical Substance Use and Addiction Research, Oslo University Hospital, Pb 4956 Nydalen, 0424 Oslo, Norway; elboka@ous-hf.no; 5The University College of Norwegian Correctional Service, 2000 Lillestrøm, Norway; 6Division of Mental Health and Addiction, Oslo University Hospital, 0424 Oslo, Norway

**Keywords:** exercise, cigarette, smokeless tobacco, substance use, health behavior

## Abstract

Exercise is increasingly understood as an important resource for people who engage in harmful substance use, including those in prison. Little is known about how inmates adopt various health behaviors during incarceration, without interventions. This cross-sectional study analyzed self-reports from 1464 inmates in Norwegian prisons in 2013–2014, compared them according to harmful substance use pre-incarceration, and explored changes in exercise and nicotine use during incarceration. Results were presented in accordance with the Strengthening the Reporting of Observational Studies in Epidemiology (STROBE) guidelines. Inmates with harmful substance use reported higher rates of smoking, smokeless tobacco, and physical inactivity pre-incarceration than inmates without harmful use. However, inmates with harmful use also exhibited more behavioral changes: they adopted exercise, ceased smoking, and adopted smokeless tobacco at higher rates during incarceration than the non-harmful group, to the extent that inmates with harmful use exercised during incarceration more. Exercise is being taken up by a significant proportion of inmates, and may in particular be a replacement behavior for substance use. However, unhealthy behaviors also begin or are maintained. If prisons were used as an arena to facilitate healthy behaviors, the public health benefits to a marginalized group such as substance-using inmates could be substantial.

## 1. Introduction

Exercise—defined as any planned, structured, or repetitive physical activities, often with the goal of increased fitness [1]—is increasingly understood as an important resource for people struggling with harmful substance use. Harmful substance use is indicated by damage to physical health, mental health, or social functioning due to drug or alcohol use, and the danger of repeated use (in addition to these harms) is the individual’s physiological adaptation to substances [2]. A recent meta-analysis reported that participation in an exercise intervention significantly increased the abstinence rates of substance use disorder patients [3]. Numerous reviews have proffered clinical and theoretical mechanisms for this efficacy, including less severe withdrawal symptoms during detoxification [3], reduction in craving [4], lessened co-morbid anxiety and depression [3,4,5], improvement of positive affect and mood [5], and reduction in stress reactivity [6]—overall reducing the chance of relapse among people in treatment and afterwards. Exercise can also serve as an important alternative behavior, an activity that takes up the time and energy otherwise consumed by substance use [7], and a majority of people with substance use disorders are interested in assistance beginning or maintaining an exercise regime [8]. 

The Norwegian health authorities [9] recommend exercise as an adjunct treatment for substance use disorders because, in addition to clinical benefits, it is cost-effective and accessible after the formal treatment system. Importantly, exercise reduces the risk of numerous preventable chronic diseases [10,11], of which people with substance use disorders already experience earlier and with more fatal consequences [12,13,14,15].

Despite this consensus, people with harmful substance use issues typically report far lower rates of exercise than the general population [8]. This is true for out-of-treatment users, inpatients [16,17], and outpatients [16], with mixed evidence from inmates [18,19]. Substance use disorders are the most common mental disorders among inmates, with pooled estimates of a 51% prevalence of drug use disorders worldwide, and a 24% prevalence of alcohol use disorder [20]. The restricted prison environment and reduced access to illicit substances may spark a need to develop new coping mechanisms and stress management techniques [21,22,23]. 

Prison can provide an ideal setting for exercise and other health behavior interventions, as the inmates’ exposure to interventions can be controlled [24,25]. Many of the barriers to exercise identified by substance users, such as a lack of time, transportation, or finances [8,26] can be easily removed in incarceration settings. Exercise interventions among inmates have increased fitness and functional cardiorespiratory capacity [27,28,29,30], improved psychological well-being [21,28,31,32], and reduced aggression [33]. Only two of these studies reported on inmates with some sort of harmful substance use: 105 inmates with “substance abuse problems” pre-incarceration, no diagnosis reported, self-reported improved physical fitness and alleviated anxiety, stress, and depression [28], and 19 inmates in methadone maintenance treatment improved strength and cardiorespiratory capacity [30]. Few studies have examined inmates’ capacities to adopt exercise without interventions. Substance users in the UK reported reductions in physical activity during the first week of incarceration [19], while cross-sectional studies from Italy and Nigeria found both positive and negative relationships, respectively, between exercise frequency and length of incarceration [34,35].

In other naturalistic studies of incarceration settings, prisons are not realizing their health-promotion potential, as weight gain [36] and unhealthy diets are commonly reported [18,37]. Furthermore, smoking is more prevalent among inmates than the general populations in 35 of 36 countries [38], which one study suggests may be because smoking is perceived as one of the few freedoms allotted to inmates [39]. A limitation of these studies is that they do not tell us how diets and smoking behaviors changed during incarceration. 

Data from the 1499 inmates participating in the Norwegian Offender Mental Health and Addiction (NorMA) study showed that 47% reported daily substance use in the six months prior to incarceration, an important indicator of potentially problematic substance use [40]. Prisons may have different health effects for substance-using and non-using inmates. In one study, inmates who used drugs before incarceration were twice as likely to self-report better health after incarceration than inmates without pre-incarceration drug use. For alcohol users, the pattern was reversed: the majority reported worsened self-rated health after incarceration [18]. 

Little research has been conducted that explores substance-using inmates’ changes in multiple health behaviors during incarceration, without interventions. It is vital to understand how the prison environment can support or hinder inmates’ health behaviors, and whether or not positive changes can be induced without the implementation of potentially costly interventions. This analysis therefore aims to answer the following questions: What is the prevalence of exercise and nicotine use among inmates with and without harmful substance use?How do exercise and nicotine use change over time?Which variables, including individual and system-related, are associated with increased exercise frequency during incarceration?

## 2. Materials and Methods

### 2.1. Setting

Norway has some of the lowest incarceration rates in the world: 74 incarcerated persons per 100,000 residents, compared to 666 per 100,000 in the United States and 114 in Canada [41]. There are 63 prison locations, about two-thirds of which are high-security; capacity ranges from 13–400 cells, with an average of 70 [42]. The Norwegian prison system is characterized by a rehabilitative perspective of incarceration, in which incarceration should help individuals re-integrate into society by equipping them with the tools to be participating citizens and to not re-offend [42,43]. The loss of liberty shall be the only deprived right, and incarcerated life should mirror life outside prison to as great as an extent possible, which means that incarceration does not have a goal of making prisoners suffer [42]. Norway also has a lower recidivism rate than most countries, estimated at 20% after two years [44].

Despite this success, the disproportionate burdens of mental and somatic multi-morbidities borne by Norwegian inmates mirror those in the rest of the world [25,45], and Norwegian inmates enter with lower education levels and higher rates of unemployment than the general population [46]. In-prison health care services do not seem to be available to all prisoners equally, according to an earlier study including half of Norwegian prisons [47]. As an example, only 13 prisons have specialized substance use disorder treatment units [42]. 

There is no national health promotion strategy for Norwegian prisons. Inmates are obliged to engage in work and daily activities, such as school and other forms of education. However, they are not obliged to exercise. Most prisons provide a gym, but exercise is usually based on the inmates’ own initiative. Only two prisons require light exercise, and only one has a complete cigarette ban. Cigarettes and smokeless tobacco are typically available for purchase at prison commissaries, although cigarette smoking has been banned in common areas in prisons since 2006 [42].

### 2.2. Study Design

This cross-sectional analysis used a large cohort study of prisoners in Norway, the Norwegian Offender Mental Health and Addiction (NorMA) study, the methodology of which has been described in Bukten et al. [48]. In the NorMA study, study investigators distributed questionnaires to 57 of 63 Norwegian prisons in 2013 and 2014. Each of the 57 prisons were visited only once, and study investigators collected self-reported survey data from 1499 inmates of high security units, low security units, and transitional facilities. The six prisons not participating housed a maximum of 179 inmates, and their non-participation was due to limited staff capacity and geographical inconvenience. There were no exclusion criteria, and inmates participated based on their interest and availability. Figure 1 displays a flow chart of participation. It is possible that an inmate could have participated early on in data collection from prison A, moved to prison B, and contributed data again from prison B without informing the research team. However, this is deemed unlikely, as among the participants who provided personal identification numbers (about half), none were duplicates. In addition, several participants who had moved prisons after participating once made this clear to the researchers.

The NorMA study received ethical approval by the Regional Committees for Medical and Health Research Ethics (2012/297), the Norwegian Social Science Data Services, and the Directorate of Norwegian Correctional Service. Each prison’s management approved all visits by researchers.

### 2.3. Measures

The questionnaire included a total of 116 items and was available in Norwegian, English, German, French, and Russian. Participants reported on both historical (pre-incarceration) and current (during incarceration) variables while filling out the questionnaire. 

#### 2.3.1. Six-Month Pre-Incarceration Variables

Participants were asked to answer the Alcohol Use Disorders Identification Test (AUDIT) and Drug Use Disorders Identification Test (DUDIT) by reflecting on their substance use in the 12 months prior to incarceration. Participants who reported either harmful alcohol use or harmful drug use were coded into the “harmful substance use” group. Harmful alcohol use was indicated by a score of ≥8 or men and ≥6 for women on the AUDIT-10, according to Norwegian guidelines [49,50]. Harmful drug use was indicated by ≥6 for men and ≥2 for women on the DUDIT [51]. Both the AUDIT and DUDIT were scored for those who answered at least five items in each, and these individuals’ missing items were replaced by the individual means, following the recommendations of Hawthorne and Elliot [52]. Those who indicated through other questions that they had no experience with substances were instructed to skip the AUDIT and DUDIT, and were coded into the ‘no harmful use’ group. 

Current psychological distress was measured by the 10-item Hopkins Symptom Checklist 10 (SCL10) on a 1–4 scale [53], in which scores ≥1.85 indicate clinical concern [54]. Due to missing data for the SCL10 (only 70.0% answered all 10 items), up to two missing values were replaced by the individual’s mean and the SCL10 mean score was calculated, capturing an additional 7.1% of participants than if only participants with zero missing were assigned mean scores. Participants also reported if they exercised, using their own definition of exercise, the amount of exercise sessions per week, if they smoked cigarettes, and if they used smokeless tobacco in the six months prior to incarceration. 

#### 2.3.2. During Incarceration Variables

Participants were then asked to report the substances they had used during their current incarceration, and overall frequency of substance use. They also reported current exercise, cigarette use, and smokeless tobacco use. The smokeless tobacco products used in Scandinavia, “snus”, have lower levels of nitrosamine than smokeless tobacco products sold in the United States and than in cigarettes [55]. Cigarette and smokeless tobacco reduction variables were created that captured changes in these behaviors over time while incarcerated, and coded on a scale from adoption to cessation: 0 = began cigarette/smokeless tobacco use, 1 = maintained, 2 = stopped, and 3 = never. Current self-rated physical health was reported on a 1–5 Likert type scale from 1 = very poor to 5 = very good, with 3 as a neutral option. 

### 2.4. Analysis

For this analysis, participants with harmful use of alcohol and/or drugs in the six months before incarceration were compared to those without harmful use. The two groups’ independent variables displayed in Table 1 were compared by chi-squares, independent sample *t*-tests, and Mann–Whitney U-tests, according to distribution and type of variable. All analyses were conducted with complete cases.

Several sets of chi-squares were then conducted to compare rates of pre-incarceration and during incarceration exercise, smokeless tobacco, and cigarette use between the two groups. To examine whether the two groups changed exercise, smokeless tobacco, or cigarette use behavior pre-incarceration to during incarceration, paired samples *t*-tests were conducted within each group to compare these dichotomous variables before and during incarceration. Changes in exercise frequency were explored further as the dependent outcome of a general linear model with repeated measures. Harmful substance use pre-incarceration was the between-subject factor and time the within-subject factor, accounting for individual variation in exercise frequency from ‘pre-incarceration’ to ‘during incarceration’. Correlations with variables previously identified as influential were tested, including age, length of incarceration, and mental distress. Neither age nor length of incarceration were correlated to current exercise frequency. Mental distress had a weak correlation (*r* = −0.21, *p* < 0.001) and was therefore not added as a covariate to the general linear model (adding mental distress would additionally have reduced the sample size by 220 due to missing values).

To explore potential factors associated with increased exercise frequency during incarceration, hierarchical multiple linear regression analyses were run separately for the harmful and non-harmful substance use groups. All bivariate relationships between exercise during incarceration and participant characteristics primarily describing current health states and behavior during incarceration (Table 1) were tested, as well as the relationships between reductions in cigarette smoking and smokeless tobacco use during incarceration. Pre-incarceration exercise frequency was entered in the first stage, followed stepwise by variables with significant bivariate relationships. The adjusted R^2^ of each adjusted model is reported. 

Results were presented in accordance with the Strengthening the Reporting of Observational Studies in Epidemiology (STROBE) guidelines [56].

## 3. Results

### 3.1. Sample Description

Women comprised 6.4% (*n* = 94) of the 1464 participants. 73.3% (*n* = 1082) reported harmful substance use pre-incarceration, of which 55.5% (*n* = 506) reported daily use of at least one substance in the year prior to incarceration. Twenty-six percent *(n* = 391) reported no harmful use before incarceration. These two groups differed in most sociodemographic and substance use variables during incarceration (Table 1). Harmful substance users were on average seven years younger and more likely to be single, were more often Nordic-born, and had lower rates of completed education and of employment or studying. They were also serving shorter sentences, and had been incarcerated for less time. Health variables during incarceration, however, did not differ; 61.8% of the harmful use group and 60.6% of the no harmful use group reported ‘good’ or ‘very good’ physical health, with 15.2% and 18.9% respectively, reporting ‘poor’ or ‘very poor’. The average mental distress score was 1.92 in the harmful use group, with 44.2% scoring above the cut-off of 1.85, indicating clinically concerning symptoms. The average score in the no harmful use was below this threshold, and only 35.9% had clinically concerning symptoms.

### 3.2. Prevalence of Exercise and Nicotine Use

Figure 2 displays prevalence rates of the various health behaviors both pre-incarceration and during incarceration. Prevalence of all three behaviors differed pre-incarceration (all *p*-values < 0.001). Harmful substance users were more likely to use both types of nicotine and less likely to exercise before being incarcerated.

During incarceration (when interviewed), exercise rates increased among both groups, and the increase in the harmful use group was so great that about two-thirds of both groups exercised, evening out the pre-incarceration difference (*p* = 0.123). Twice as many harmful substance users used smokeless tobacco than the no harmful use group (*p* < 0.001). Three quarters of harmful substance users smoked cigarettes, while only one half of the no harmful substance use group smoked (*p* < 0.001). 

### 3.3. Changes in Exercise and Nicotine Use during Incarceration

Figure 2 also displays changes in these behaviors during incarceration. Rates of exercise increased for both groups. Cigarette use decreased only among harmful substance users, while smokeless tobacco rates increased significantly for both groups. 

The non-harmful use group reported pre-incarceration exercise at higher frequencies (1.9 days/week) than the harmful use group (1.4 days/week); Figure 3. During incarceration, the cohort as a whole did not increase exercise frequency, with a non-significant main effect of time for the entire sample (F (1, 1245) = 97.2, *p* = 0.072). This increase was larger for the harmful substance use group, which increased from 1.4 to 2.7 days/week (significant interaction effect of group time, F (1, 1245) = 20.4, *p* = 0.016).

### 3.4. Factors Related to Increased Exercise Frequency 

Table 2 displays the results of the regression model explaining variance in exercise frequency for the harmful substance use group. All variables with significant bivariate relationships to exercise frequency were requested stepwise, but the following variables were not included in the final models: mental distress, reduction in cigarettes, reduction in smokeless tobacco, anabolic androgenic steroid use during incarceration, Nordic birth, and months of sentence. The regression equation was significant, F (4, 6) = 35.8, *p* < 0.001. Five variables explained 18.7% of the variance in exercise frequency during incarceration, and the most explanatory variable was self-rated physical health (β = 0.294). Pre-incarceration exercise frequency, added with forced entry into the model, was the next most important predictor (β = 0.180), while older age (β = −0.146) was a negative predictor. Having at least a secondary school education was also a positive predictor of higher exercise frequency (β = 0.101). 

For the non-harmful use group (Table 3), three variables explained 23.7% of the variance of exercise frequency during incarceration, in a significant regression equation (F (3, 249) = 25.8, *p* < 0.001). Pre-incarceration exercise frequency was more important than for the harmful group (β = 0.355), while satisfaction with physical health was nearly as important, and explained a stable amount of variance with the inclusion of each new predictor variable (β = 0.274). Nordic birth was the only other positive predictor (β = 0.140). The following variables had significant bivariate relationships but were not included in the adjusted models: age, gender, smokeless tobacco reduction, and cigarette reduction.

## 4. Discussion

Inmates with harmful substance use entered prison with higher rates of negative health behaviors than inmates without problematic use, including 81.3% who smoked cigarettes, 61.3% who were physically inactive, and 26.0% who used smokeless tobacco. However, inmates with harmful use also exhibited more behavioral changes during incarceration: they adopted exercise, ceased smoking, and adopted smokeless tobacco at higher rates during incarceration than the non-harmful group. The non-harmful group also adopted exercise and smokeless tobacco, but exhibited no changes in cigarette use during incarceration. 

The significant positive change in exercise behavior among inmates with harmful substance use, both in the amount of inmates beginning to exercise and in frequency, suggests that exercise may be an important replacement behavior for substance use under incarceration. This replacement mechanism could be a de facto replacement, as inmates have reduced access to substances: 22.7% of this group reported substance use at least four times during incarceration, which is a clear decline from the 55.5% daily use reported pre-incarceration. Exercise likely also induces neurological adaptations in reward-, inhibition-, and stress-related systems that directly counter and compete with the effects of substances in these same systems [57,58].

Pre-incarceration exercise was found to predict current exercise and this was expected, as exercise is a behavior in which within-person variation is decisive in predicting change. The low correlation of current mental distress to current exercise frequency is promising, as it suggests that mental health concerns need not be assumed prohibitive to inmate exercise. Muller and Clausen [59] were similarly able to engage residential substance use disorder patients with the highest mental distress in a pilot group exercise program. Age negatively predicted exercise frequency for harmful substance users, but was unrelated to non-harmful users’ exercise. Manocci et al. [35] also found exercise frequency in Italian prisons to be negatively related to age, and positively related to physical health-related quality of life and non-Italian nationality, with no relationship between exercise and amount of cigarettes or education level. Furthermore, sentence duration was not an important predictor of exercise frequency in our analysis, in line with Manocci et al.’s findings but not those of Olaitan et al. [34] in Nigeria. 

Previous longitudinal, population-based studies have found exercise uptake to result in improved self-rated health [60], and exercise reductions in worsened self-rated health [61]. Similarly, we found strong correlations of current exercise and current self-rated health among both inmate groups. As with non-exercisers, inmates with poor health could be targeted as having potentially more to gain. Exercise cannot change incarceration itself or elements such as overcrowding or a lack of healthcare, but it could change the experience of incarceration, such as by providing a sense of autonomy, a challenge to boredom, and a relief against stress, elements which have been identified as reasons for the poor health of inmates [62]. Helping inmates exercise can also be seen as a way to equip them with an anti-depressive and stress-reducing tool that they can continue to use post-release [63,64]. One-third of inmates in a previous American study experienced an increase in depression and stress after release, probably reflecting the environmental stressors of community living and the difficulties of post-release reintegration [65]. Exercise could thus be a tool used not only during, but also after, incarceration. 

It is important to emphasize that exercise should not be uncritically assumed a positive behavior. Exercise will necessarily increase the risk for exercise-related physical injuries, for example, and health screening may be necessary to identify prisoners with circulatory or heart problems who would benefit more from low-intensity or otherwise modified exercise [66]. Meek and Lewis also speculate that some prisoners with low self-esteem, eating disorders, or body disorders may be more predisposed to anabolic steroid use if they begin exercising. Steroid use is associated with a range of adverse health consequences, including cardiovascular effects and mental health problems [67,68,69,70], and lifetime anabolic steroid use is already found to be many times higher among prisoners [71,72] than the general population [73]. These individuals could greatly benefit from learning that the positive effects of exercise can be reached without steroids. 

While many inmates were able to adopt exercise without structured interventions, nicotine use increased for the cohort as a whole. Inmates with harmful substance use reduced their rates of smoking during incarceration, yet smoking rates remained quite high, as has been reported internationally [38]. Three-quarters of the harmful use group continued smoking when interviewed, as did half of the non-harmful substance use group, and both of the groups increased rates of smokeless tobacco use. This suggests that various types of nicotine use among inmates will not be reduced in the absence of cessation programs (or among harmful substance users, in the absence of the ability to exercise). It is also likely that some inmates are substituting smokeless tobacco for cigarettes, or using smokeless tobacco to reduce their cigarette use. A recent meta-analysis of 85 articles concluded that complete smoking bans can successfully reduce smoking rates during incarceration, but only smoking cessation programs have effects that last post-incarceration [38]. Makris et al. [74] reported that simply the establishment of a cessation program may provide motivation to quit. At the same time, inmates are able to understand the health risks of smoking and still choose to smoke: van den Berg et al. [39] found that in a prison with a complete smoking ban, prisoners who perceived smoking as an expression of freedom were more likely to plan to resume smoking upon release compared to those who without such an association. Yet after release, all prisoners reduced their average amount of cigarettes by half—again supporting the establishment of cessation programs, rather than bans, and highlighting the need to understand why inmates engage in various health behaviors.

### 4.1. Limitations and Strengths

Some of the limitations of this analysis arise from the data collected in the questionnaire, most obviously that causation cannot be concluded from cross-sectional data. The cigarette and smokeless tobacco variables were dichotomous, therefore we were not able to differentiate between casual smokers or smokeless tobacco users from daily users. Similarly, lacking a standard definition of exercise, participants’ reports may not represent recommended amounts of exercise. Somatic health problems were outside the scope of the NorMA study, and while the self-rated health question was likely a strong proxy for health limitations, it was not possible to identify whether certain problems—for example, cardiovascular disease or obesity—particularly inhibited the adoption of exercise.

The intention of this study was to collect participant-reported information. It is difficult to predict whether a population will over- or under-report current exercise, according to a meta-analysis [75], but self-reports of historical exercise, from 24 h to 10 years earlier, have been reported to be valid in numerous countries, although currently overweight individuals may over-report historical exercise [76,77,78]. If certain groups over-reported pre-incarceration exercise, then exercise may have been adopted during incarceration to an even greater extent than reported here. In general, participants in the NorMA study were representative of the national prison population in terms of gender, citizenship, and country of birth [48].

### 4.2. Clinical Implications

An array of earlier literature has revealed the large health burden of substance users, such as early onset cardiovascular disease and hypertension [79,80]. The negative health behaviors documented in this paper can, if modified, reduce many future health risks. Our findings strengthen the argument for prisons to enable exercise as a public health intervention, with demonstrated benefits including improved physical and mental health [21,27,28,29,30,31,32], and reduced aggression [33], and inmate-identified benefits in one review including improved self-esteem, confidence, and the construction of a new identity [81]. Incarceration should be seen as an opportunity for positive behavior change, and it is encouraging that inmates with pre-incarceration harmful substance use seem particularly able to adopt such changes. Given that inmates’ need for substance treatment far outpaces access to treatment, actively facilitating exercise among inmates even without instituting formal interventions—e.g., by increasing the amount of time available for exercise, preventing inmate exclusion from facilities by other inmates, or resisting the revocation of exercise privileges as a punishment measure—could be a cost-effective method to provide substance-using inmates with a healthy alternative to substances. It may be particularly important to facilitate exercise among non-exercisers entering prison, as this group may not have the health knowledge or self-efficacy to begin independently, or may have health needs that require tailored or facilitative programs [66]. Prison staff should be aware of lifetime steroid use and current steroid risk, such as harmful substance use and being underweight [82], and future qualitative research among inmates should aim to understand the meanings of steroid use, exercise, and other health behaviors explored in this article.

## 5. Conclusions

Prisons are environments housing marginalized populations with large health burdens, and they should be seen as public health opportunities. Overall, our results support the health-promoting potential and necessity of prisons: inmates entered with high rates of negative health behaviors, and substance-using inmates were particularly burdened. Behavioral change was common during incarceration, specifically the adoption of exercise among inmates with problematic substance use, which may have been related to being deprived of substances. Future research should explore the meaning of exercise for inmates and its potential to act as a substitute behavior for substance use, while prisons themselves should facilitate a range of exercise options, and institute nicotine cessation programs. 

## Figures and Tables

**Figure 1 ijerph-15-02663-f001:**
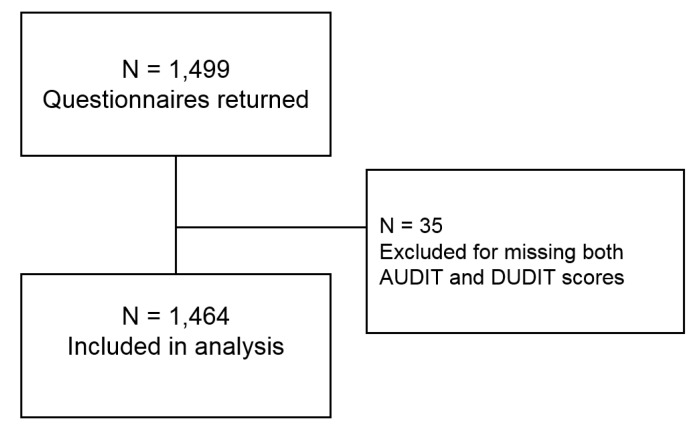
Flow chart of participation in the Norwegian Offender Mental Health and Addiction (NorMA) study. AUDIT: Alcohol Use Disorders Identification Test. DUDIT: Drug Use Disorders Identification Test.

**Figure 2 ijerph-15-02663-f002:**
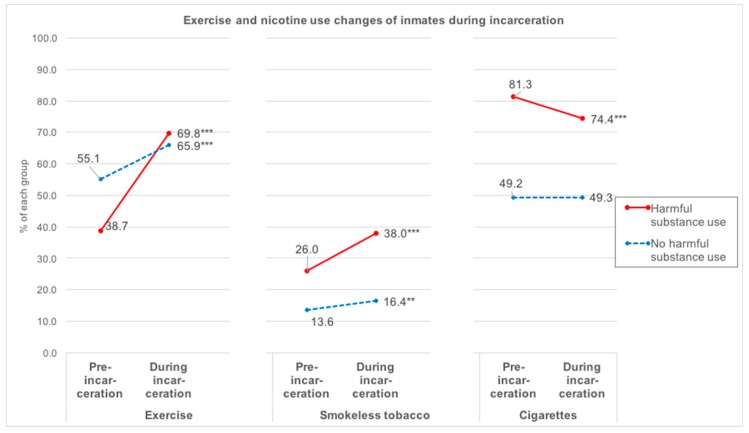
Exercise and nicotine use changes of inmates during incarceration. Within-group differences over time indicated by: * *p* < 0.05, ** *p* < 0.01, *** *p* < 0.001.

**Figure 3 ijerph-15-02663-f003:**
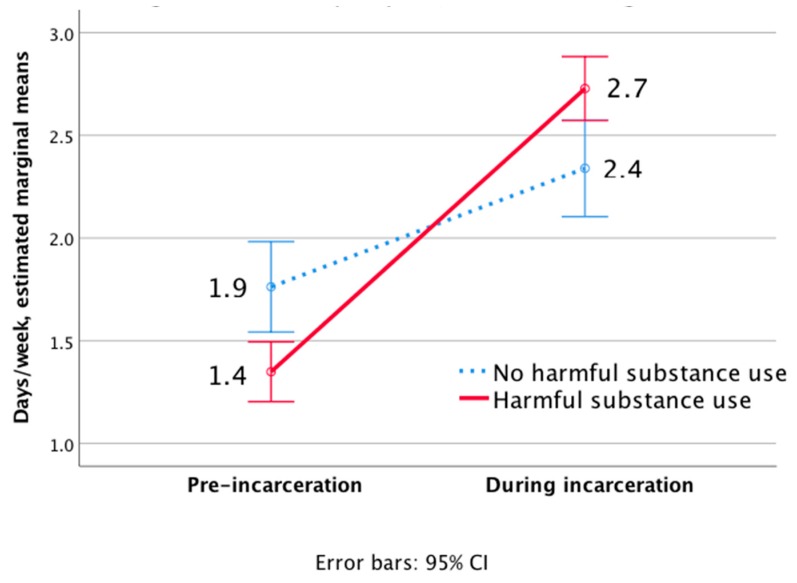
Changes in exercise frequency (sessions/week) of inmates during incarceration.

**Table 1 ijerph-15-02663-t001:** Description of 1464 inmates, by pre-incarceration harmful substance use.

Variables	Harmful Substance Use, *n* = 1073	No Harmful Substance Use, *n* = 391	Test Statistic	*n* Missing
	*n* (%)	*n* (%)		
Women	65 (6.1)	29 (7.4)	*X*^2^_(1)_ = 0.88	0
Age (mean, SD)	32.8 (*10.1*)	39.7 (*12.8*)	*z* = −9.04 ***	123
Nordic-born	833 (79.3)	203 (54.9)	*X*^2^_(1)_ = 80.56 **	44
Secondary school or more	616 (58.1)	287 (74.5)	*X*^2^_(1)_ = 34.52 ***	18
Single	785 (73.8)	199 (52.1)	*X*^2^_(1)_ = 61.20 ***	18
Working or studying before incarceration	465 (43.3)	277 (70.8)	*X*^2^_(1)_ = 90.83 ***	0
Incarceration				
Months of sentence (median, IQR)	14.0 (28.0)	27.0 (42.2)	*z* = −4.15 ***	274
Months served (median, IQR)	5.0 (11.3)	6.3 (19.3)	*z* = −2.67 **	112
Security level			*X*^2^_(2)_ = 4.48	12
High-security	614 (57.5)	215 (55.8)		
Low-security	420 (39.0)	149 (38.7)		
Transitional housing	34 (3.2)	21 (5.5)		
Health during incarceration				
SCL10 score (mean, SD)	1.92 (0.79)	1.77 (0.75)	*t*_(1164)_ = −3.28 **	320
Self-rated physical health			*X*^2^_(4)_ = 9.15	21
Very poor	53 (5.0)	22 (5.7)		
Poor	108 (10.2)	51 (13.2)		
Neither poor nor good	245 (23.2)	79 (20.5)		
Good	421 (39.8)	130 (33.8)		
Very good	231 (21.8)	103 (26.8)		
Substance and unprescribed medication use during incarceration				
Frequency			*z* = −6.92 ***	65
None	541 (62.1)	119 (93.0)		
Once	60 (6.9)	4 (3.1)		
2–3 times	72 (6.7)	2 (1.6)		
4+ times	198 (22.7)	3 (2.3)		
Substances used				351
Cannabis	233 (21.7)	3 (0.8)	*X*^2^_(1)_ = 93.9 6 ***	
OMT medicine, e.g., methadone	168 (15.7)	2 (0.5)	*X*^2^_(1)_ = 62.56 ***	
Benzodiazepines or sedatives	123 (11.5)	1 (0.3)	*X*^2^_(1)_ = 45.40 ***	
Meth/amphetamines	68 (6.3)	1 (0.3)	*X*^2^_(1)_ = 25.83 ***	
Synthetic cannabis	43 (4.0)	0 (0)	*X*^2^_(1)_ = 15.80 *	
Methylphenidate ^a^, e.g., Ritalin	39 (3.6)	0 (0)	*X*^2^_(1)_ = 14.29 ***	
Heroin	29 (2.7)	1 (0.3)	*X*^2^_(1)_ = 8.33 **	
Cocaine	26 (2.4)	0 (0)	*X*^2^_(1)_ = 9.44 **	
Gamma-hydroxybutyric acid (GHB)	20 (1.9)	0 (0)	*X*^2^_(1)_ = 7.23 **	
Anabolic steroids	17 (1.6)	0 (0)	*X*^2^_(1)_ = 6.14 *	
Inhalants	11 (1.0)	0 (0)	*X*^2^_(1)_ = 3.95 *	
Ecstasy	9 (0.8)	0 (0)	*X*^2^_(1)_ = 3.23	
LSD, PCP, or ketamine	7 (0.7)	0 (0)	*X*^2^_(1)_ = 2.51	

Inmates with harmful substance use before incarceration were younger, more likely to be single, had lower rates of completed education and of employment or studying, and had a higher prevalence of mental distress compared to inmates without harmful substance use. *X*^2^ refers to chi-square for nominal dependent variables. *z* refers to Mann–Whitney U test for nonparametric variables. OMT: opioid maintenance treatment. ^a^ A central nervous system stimulant prescribed for attention-deficit hyperactivity disorder. * *p* < 0.05, ** *p* < 0.01, *** *p* < 0.001.

**Table 2 ijerph-15-02663-t002:** Adjusted models explaining variance in exercise frequency during incarceration of inmates with harmful substance use (*n* = 605).

Variables	Model			
	1	2	3	4
(Constant)	---	---	---	---
Pre-incarceration exercise frequency	0.247 ***	0.202 ***	0.187 ***	0.180 ***
Self-rated physical health		0.326 ***	0.297 ***	0.294 ***
Age			−0.136 **	−0.146 ***
Secondary school education or higher				0.101 **
Adjusted R^2^	5.9%	16.3%	17.8%	17.8%

** *p* < 0.01, *** *p* < 0.001.

**Table 3 ijerph-15-02663-t003:** Adjusted models explaining variance in exercise frequency during incarceration of inmates with no harmful substance use (*n* = 253).

Variables	Model		
	1	2	3
(Constant)	---	---	---
Pre-incarceration exercise frequency	0.388 ***	0.358 ***	0.355 ***
Self-rated physical health		0.261 ***	0.274 ***
Nordic birth			0.140 *
Adjusted R^2^	14.7%	21.7%	23.7%

* *p* < 0.05, *** *p* < 0.001.

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
