# Peer review of "Inmates with Harmful Substance Use Increase Both Exercise and Nicotine Use Under Incarceration"

_ijerph, 2018, doi:10.3390/ijerph15122663_

Round 1
Reviewer 1 Report
Review report IJERPH: Inmates with harmful substance use increase both exercise and nicotine use under incarceration
The authors examined in their paper changes in exercise and tobacco use behaviors among 1,468 inmates in Norwegian prisons. Furthermore, they made a difference between inmates who reported substance use before incarceration versus inmates who reported no substance use. Data were collected in 2013 and 2014 using questionnaires. The authors concluded that all inmates increased their exercise behavior, but this change was far stronger among inmates who had used substances before incarceration. Furthermore, a significant number of respondents who had used substances quit smoking during incarceration.
The topic addressed, i.e. examining behavior change without any form of intervention, is an interesting and relevant one, in particular in the context of relapse prevention of substance use. One of the major strengths of the paper is that it is very well written and the results are described in a very precise and detailed way. Furthermore, the use of figures is very supporting and the implications of the findings are clearly defined. However, I also have some questions and suggestions for the authors:
Title: is this correct? The title now states that inmates increased their nicotine use, but they decreased it or not?
Abstract: I would recommend to indicate that this is a repeated cross-sectional study and also add the data collection period. The reader assumes now that it is a longitudinal study.
Abstract: the authors describe in the results the baseline associations, but this is not described before. The reader wonders here why they describe smoking, use of smokeless tobacco and physical inactivity pre-incarceration instead of during incarceration as announced before.
Introduction: line 44: the heading is accidentally included again.
Introduction: I would suggest to highlight the relevance of this study even more. For example, interventions can cost a lot of time and money, and if the authors would find that such interventions are not even necessary for comparable effects, this would be a very cost-effective respectively cost-saving result, especially concerning smoking (for physical activity, there might be some facilities etc. needed).
Methods: so far, it was not clear that this is a repeated cross-sectional study. However, some respondents could have been measured twice I think. How were they treated in the study?
Methods: line 138: an "f" is missing (For men).
Methods: were the questions about exercise validated? Or was only asked about exercise in general without the type of exercise? This is especially important in the analyses about frequency of exercising. For example, did this include walking for 10 minutes or was that sporting for at least an hour?
Methods: the authors describe that they removed ‘mental distress’ from the analyses. Does that mean that the other two variables (age and length of incarceration) had no relationship and were also not added or had a strong relationship and were added?
Results: what is meant with those who had reported no harmful use, but daily use (line 198)?
Results: Figure 3 shows only changes in exercising I think (not smoking behavior) (line 221).
Results: line 225: I think this should be Figure 3 instead of Figure 2.
Results: steroid use was not described before in the methods. Please describe the full model that was tested.
Results: why did the authors chose to focus only on frequency of exercising? Why were, for example, no predictors of smoking cessation examined?
Results: I would recommend to report in the tables which other variables were included in the model, in addition to the significant ones.
Discussion: I would not only focus on inmates with a substance use history because this reads as if the other group did not improve their behaviors. Of course, this is the larger and more at-risk group, but actually all inmates changed their behavior.
Conclusion: the first sentence reads a little overstated, as if prisons are a positive thing. I think what the authors mean is that if inmates are there anyway, they can also use this time to improve several health behaviors. But of course, this would theoretically also be possible outside the prison.
Conclusion: why is the focus of the conclusion only on exercise and not smoking behavior?
Conclusion: the authors recommend that nicotine cessation programs should be implemented, while I think they weaken their own findings with this. Of course, programs would probably even enhance that effect, but also without such programs, inmates often quit or reduce smoking as was shown in their study.
Author Response
The authors examined in their paper changes in exercise and tobacco use behaviors among 1,468 inmates in Norwegian prisons. Furthermore, they made a difference between inmates who reported substance use before incarceration versus inmates who reported no substance use. Data were collected in 2013 and 2014 using questionnaires. The authors concluded that all inmates increased their exercise behavior, but this change was far stronger among inmates who had used substances before incarceration. Furthermore, a significant number of respondents who had used substances quit smoking during incarceration.
The topic addressed, i.e. examining behavior change without any form of intervention, is an interesting and relevant one, in particular in the context of relapse prevention of substance use. One of the major strengths of the paper is that it is very well written and the results are described in a very precise and detailed way. Furthermore, the use of figures is very supporting and the implications of the findings are clearly defined. However, I also have some questions and suggestions for the authors:
Title: is this correct? The title now states that inmates increased their nicotine use, but they decreased it or not?
Response: When considering both smokeless tobacco and cigarettes as nicotine use, inmates as a whole increased nicotine use. We have added more to the discussion about nicotine use, so hopefully our explanation of inmates´ nicotine use is also clearer.
Abstract: I would recommend to indicate that this is a repeated cross-sectional study and also add the data collection period. The reader assumes now that it is a longitudinal study.
Response: Added.
Abstract: the authors describe in the results the baseline associations, but this is not described before. The reader wonders here why they describe smoking, use of smokeless tobacco and physical inactivity pre-incarceration instead of during incarceration as announced before.
Response: We have expanded this sentence so that we mention current behaviors in addition to baseline behaviors and changes: “However, inmates with harmful use also exhibited more behavioral changes: they adopted exercise, ceased smoking, and adopted smokeless tobacco at higher rates during incarceration than the non-harmful group, to the extent that inmates with harmful use exercised during incarceration more.”
Introduction: line 44: the heading is accidentally included again.
Response: Removed.
Introduction: I would suggest to highlight the relevance of this study even more. For example, interventions can cost a lot of time and money, and if the authors would find that such interventions are not even necessary for comparable effects, this would be a very cost-effective respectively cost-saving result, especially concerning smoking (for physical activity, there might be some facilities etc. needed).
Response: Thank you for this suggestion. We have added the following sentence, before we introduce the paper’s three aims: “It is vital to understand how the prison environment can support or hinder inmates’ health behaviors, and whether or not positive changes can be induced without the implementation of potentially costly interventions.”
Methods: so far, it was not clear that this is a repeated cross-sectional study. However, some respondents could have been measured twice I think. How were they treated in the study?
Response: We have clarified that this is a cross-sectional study that collected current and historical data from respondents, at one point in time: “Participants reported on both historical (pre-incarceration) and current (during incarceration) variables while filling out the questionnaire… It is possible that an inmate could have participated early on in data collection from prison A, moved to prison B, and contributed data again from prison B. However, this is deemed unlikely, as among the participants who provided personal identification numbers (about half), none were duplicates.”
Methods: line 138: an "f" is missing (For men).
Response: Corrected.
Methods: were the questions about exercise validated? Or was only asked about exercise in general without the type of exercise? This is especially important in the analyses about frequency of exercising. For example, did this include walking for 10 minutes or was that sporting for at least an hour?
Response: No validated measure was used for exercise, which we have clarified: “Participants also reported if they exercised, using their own definition of exercise…”
Methods: the authors describe that they removed ‘mental distress’ from the analyses. Does that mean that the other two variables (age and length of incarceration) had no relationship and were also not added or had a strong relationship and were added?
Response: Age and length of incarceration had no significant relationships to exercise frequency, and were not added; mental distress was not added due to its weak correlation and high missing values. We have clarified this: “Neither age nor length of incarceration were correlated to current exercise frequency. Mental distress had a weak correlation (r= -.21, p<0.001) and was therefore not added as a covariate to the general linear model adding it to the general linear model (adding mental distress would additionally have reduced the sample size by 220 due to missing values).”
Results: what is meant with those who had reported no harmful use, but daily use (line 198)?
Response: We have removed this sentence.
Results: Figure 3 shows only changes in exercising I think (not smoking behavior) (line 221).
Response: Corrected.
Results: line 225: I think this should be Figure 3 instead of Figure 2.
Response: Corrected.
Results: why did the authors chose to focus only on frequency of exercising? Why were, for example, no predictors of smoking cessation examined?
Response: Unfortunately the questionnaire did not contain reliable items that measured the “frequency” or “intensity” of cigarette or smokeless tobacco use, such as those asking for the average amount of cigarettes smoked or the length of time a container of smokeless tobacco lasts. This prohibited us from being able to distinguish between participants who may have drastically reduced their cigarette use under incarceration, but still smoked, or between participants who entered prison as casual smokers and became heavy smokers. In other words, the binary yes/no categories of cigarette use and smokeless tobacco at two time points were seen as less useful than the exercise intensity variables we had, as measured through frequency.
Results: I would recommend to report in the tables which other variables were included in the model, in addition to the significant ones.
Response: Thank you for this suggestion. This information had been reported under each relevant table, which evidently was not clear enough. We have now moved the description of the full models above each table.
Results: steroid use was not described before in the methods. Please describe the full model that was tested.
Response: We have added a description of all variables tested.
Discussion: I would not only focus on inmates with a substance use history because this reads as if the other group did not improve their behaviors. Of course, this is the larger and more at-risk group, but actually all inmates changed their behavior.
Response: Correct, with the exception that the no-harmful use group did not change cigarette use. We have added a sentence to the first paragraph of the discussion that clarifies this: “The non-harmful group also adopted exercise and smokeless tobacco, but exhibited no changes in cigarette use during incarceration.”
Conclusion: the first sentence reads a little overstated, as if prisons are a positive thing. I think what the authors mean is that if inmates are there anyway, they can also use this time to improve several health behaviors. But of course, this would theoretically also be possible outside the prison.
Response: We have changed the conclusion so that it begins with this sentence, rather than implying prisons are purely positive situations: “Prisons are environments housing marginalized populations with large health burdens, and they should be seen as public health opportunities.”
Conclusion: why is the focus of the conclusion only on exercise and not smoking behavior?
Response:. We have added in a paragraph on smoking behavior: “While many inmates were able to adopt exercise without structured interventions, tobacco use increased for the cohort as a whole. Inmates with harmful substance use reduced their rates of smoking during incarceration, yet smoking rates remained quite high, as has been reported internationally (Spaulding, Eldridge et al. 2018). Three-quarters of the harmful use group continued smoking when interviewed, as did half of the non-harmful substance use group, and both of the groups increased rates of smokeless tobacco use. This suggests that various types of tobacco use among inmates will not be reduced in the absence of cessation programs (or among harmful substance users, in the absence of the ability to exercise). It is also likely that some inmates are substituting smokeless tobacco for cigarettes, or using smokeless tobacco to reduce their cigarette use. A recent meta-analysis of 85 articles concluded that complete smoking bans can successfully reduce smoking rates during incarceration, but only smoking cessation programs have effects that last post-incarceration (Spaulding, Eldridge et al. 2018). Makris, Gourgoulianis et al. (2012)reported that simply the establishment of a cessation program may provide motivation to quit. At the same time, inmates are able to understand the health risks of smoking and still choose to smoke: van den Berg, Bock et al. (2014)found that in a prison with a complete smoking ban, prisoners who perceived smoking as an expression of freedom were more likely to plan to resume smoking upon release compared to those who without such an association. Yet after release, all prisoners reduced their average amount of cigarettes by half – again supporting the establishment of cessation programs, rather than bans, and highlighting the need to understand why inmates engage in various health behaviors.”
Conclusion: the authors recommend that nicotine cessation programs should be implemented, while I think they weaken their own findings with this. Of course, programs would probably even enhance that effect, but also without such programs, inmates often quit or reduce smoking as was shown in their study.
Response: In our new paragraph on smoking behavior, we have added a discussion about nicotine cessation programs.
Reviewer 2 Report
This is a novel and interesting area of research, one which is underrepresented in the literature. Equity in research is essential and reaching an audience like incarcerated individuals to improve their health may help to address this. As well, producing behaviour change at a level which, once back in community, may be extended to improve their health and lifestyle is important to reduce social isolation.
1. Introduction- Perhaps an editorial error- remove text “1. Introduction”
2. Measures- p4, line 140. “by ≥6 or men” change or to for.
3. Is harmful use people who have both harmful alcohol use and harmful drug use? If so make this clear earlier, perhaps in the measures section where discussing AUDIT and DUDIT.
4. Line 146, aas there any difference between those scores which had imputed data versus those which had all available data?
5. Results, line 197, starts sentence with number “26.3%”, suggest rephrasing with words.
6. Table 1, Anabolic steroid line, column two, “0(0=” replace ‘=’ with ‘)’
7. Line 243, states 5 variables explained 22.7% of the variance however table three only has 3 variables in it?
8. Table 3 is not referenced in the text?
9. Figure 2. Smokeless tobacco use went up, cigarette consumption down, is it able to be determined the number of tobacco smokers pre incarceration who switched to smoke-less tobacco during their incarceration? It may not be that individuals are quitting smoking but more so switching mechanisms for nicotine delivery (smoking cessation is mentioned on line 254)
10. Was the rate of exercise increase as high for current tobacco users? Given that they have access to their substance of choice, there may be a within group difference again?
11. Is there any information around exercise programmes being delivered to inmates and the impact of these on health outcomes? Also the benefits of exercise on their life beyond prison (social isolation, reduce health inequities, etc). Could be included in discussion though not necessary
Author Response
1. Introduction- Perhaps an editorial error- remove text “1. Introduction”
Response: Removed.
2. Measures- p4, line 140. “by ≥6 or men” change or to for.
Response: Corrected.
3. Is harmful use people who have both harmful alcohol use and harmful drug use? If so make this clear earlier, perhaps in the measures section where discussing AUDIT and DUDIT.
Response: Thank you; we have clarified this in a new sentence: “Participants who reported either harmful alcohol use or harmful drug use were coded into the “harmful substance use” group.”
4. Line 146, was there any difference between those scores which had imputed data versus those which had all available data?
Response: Yes – the 126 participants who were missing one or two SCL10 items and therefore received imputed values had average SCL10 scores greater than the non-imputed group (t= -3.9, p<0.001), indicating higher distress. The non-imputed group’s mean was 1.85, exactly the threshold for the clinically concerning distress, whereas the non-imputed group’s mean score was 2.15, also indicating distress.
5. Results, line 197, starts sentence with number “26.3%”, suggest rephrasing with words.
Response: Corrected.
6. Table 1, Anabolic steroid line, column two, “0(0=” replace ‘=’ with ‘)’
Response: Thank you; corrected.
7. Line 243, states 5 variables explained 22.7% of the variance however table three only has 3 variables in it?
Response: Corrected.
8. Table 3 is not referenced in the text?
Response: Added the reference in the text above.
9. Figure 2. Smokeless tobacco use went up, cigarette consumption down, is it able to be determined the number of tobacco smokers pre incarceration who switched to smoke-less tobacco during their incarceration? It may not be that individuals are quitting smoking but more so switching mechanisms for nicotine delivery (smoking cessation is mentioned on line 254)
Response: Thank you for pointing this out. There were 150 participants who stopped smoking and 201 who began using smokeless tobacco. Among the 150 who stopped smoking, 36% began using smokeless tobacco, 42% did not start using smokeless tobacco, 19% maintained smokeless tobacco, and only 3% stopped using smokeless tobacco. So smokeless tobacco as a substitution behavior is seen only among about one third of participants.
10. Was the rate of exercise increase as high for current tobacco users? Given that they have access to their substance of choice, there may be a within group difference again?
Response: In both of the regression analyses (for the harmful and non-harmful substance groups), a higher exercise frequency was associated at the bivariate level with reductions in cigarette and smokeless tobacco use. You are correct that current non-users of tobacco exercised at lower frequency. However, neither of the tobacco variables were significant predictors of exercise frequency in the adjusted models.
11. Is there any information around exercise programmes being delivered to inmates and the impact of these on health outcomes? Also the benefits of exercise on their life beyond prison (social isolation, reduce health inequities, etc). Could be included in discussion though not necessary
Response: Thank you for this suggestion. We have added a sentence in “clinical implications” in which we review the benefits of exercise for inmates: “Our findings strengthen the argument for prisons to enable exercise as a public health intervention, with demonstrated benefits including improved physical and mental health (Genovese, Libbus et al. 1995, Nelson, Specian et al. 2006, Perez-Moreno, Camara-Sanchez et al. 2007, Cashin, Potter et al. 2008, Buckaloo, Krug et al. 2009, Battaglia, di Cagno et al. 2013, Battaglia, di Cagno et al. 2015), and reduced aggression (Wagner, McBride et al. 1999), and inmate-identified benefits in one review including improved self-esteem, confidence, and the construction of a new identity (Martinez-Merino, Martín-González et al. 2017).